# A Unified Framework for Generalized Hierarchical Diffusion via Simplicial Complexes

## Abstract

In this paper, we propose a unified framework for hierarchical diffusion via simplicial complexes (HDSC), which enables adaptive diffusion across different levels of simplicial complexes, including nodes, edges, and triangles. To ensure the accuracy and consistency of information transmission during the diffusion process, we investigate topological consistency constraints, achieving efficient coupling between structures at various levels. Additionally, by introducing a time-dependent topological memory mechanism, we further enhance the smoothness and coherence of global information flow, enabling features at different levels to diffuse cooperatively throughout the entire graph structure. Experimental results demonstrate that HDSC exhibits significant performance advantages over traditional methods. Furthermore, as the complexity and dimensionality of the graph increase, HDSC continues to maintain its superiority, effectively avoiding the phenomenon of node feature homogenization.

## 1 Introduction

Graph structures are capable of capturing complex relationships between entities in an intuitive manner, making them widely applicable across a variety of real-world scenarios, including transportation networks, social networks, and biomolecular networks. To better handle these intricate graph structures, graph neural networks (GNNs) have emerged as an effective tool for processing graph-based data. GNNs leverage message-passing mechanisms to propagate information across graph structures, enabling nodes to update their representations based on the information from their neighboring nodes (Kipf & Welling, 2017; Veličković et al., 2018; Hamilton et al., 2017). GNNs have been extensively applied to tasks such as node classification (Wu et al., 2019; Shi et al., 2021), link prediction (Zhang et al., 2023; Liu et al., 2023), graph classification (Luo et al., 2023; Wei et al., 2023), and graph generation (Kong et al., 2023; Cong et al., 2023), achieving significant success across various domains.

Despite the success of GNNs in handling graph data, limitations in their information propagation process have gradually surfaced, In particular, there is the issue of over-smoothing, where nodes lose their distinctiveness as their features become homogenized (Qureshi et al., 2023; Giraldo et al., 2023; Chen et al., 2023). To address these challenges, diffusion equations have been introduced into graph structures, offering a continuous perspective for modeling information propagation (Atwood & Towsley, 2016; Zhao et al., 2021). Diffusion models are originally employed to describe the spread of heat or particles in physical systems, and are based on the principle that information diffuses from regions of high concentration to low concentration, until eventually reaching the equilibrium (Paul et al., 2014). In the context of graph structures, diffusion models simulate feature propagation between nodes and provide a continuous-time interpretation of message-passing mechanisms of GNNs (Chamberlain et al., 2021; Thorpe et al., 2022). Existing studies suggest that diffusion equations can provide a unified theoretical framework for GNNs, bridging the gap between discrete graph structures and continuous dynamic processes (Gasteiger et al., 2019; Li et al., 2024; Liu et al., 2024). For example, Chamberlain et al. (2021) formalizes graph learning as a continuous diffusion process, viewing GNNs as discrete approximations of underlying partial differential equations (PDEs), systematically addressing common issues in deep GNNs such as the difficulty of training deep networks and the over-smoothing of node features. Thorpe et al. (2022) extends this idea by proposing a graph neural diffusion framework with a source term, constructing a continuous deep graph learning architecture particularly suited for low-label-rate scenarios with few labeled

nodes. Gasteiger et al. (2019) introduces generalized graph diffusion (e.g., heat kernel and personalized PageRank) into graph convolution, incorporating information from multi-hop neighbors to mitigate the over-smoothing problem. Liu et al. (2024) further builds on (Gasteiger et al., 2019) by incorporating adversarial perturbation mechanisms through min-max optimization, enhancing the model's robustness against adversarial attacks and noise in graph structures.

However, despite the strong potential demonstrated by the integration of diffusion equations and GNNs, most existing works are primarily focused on handling node-to-edge relationships, lacking a systematic extension to higher-order structures within graphs. Current graph diffusion methods face several limitations:

(1) Limiting to modeling low-order relationships: Most existing graph diffusion convolution methods (Chamberlain et al., 2021; Thorpe et al., 2022; Gasteiger et al., 2019; Li et al., 2024; Liu et al., 2024) primarily focus on low-order relationships between nodes and edges, overlooking the geometric and topological information embedded in higher-order structures (e.g., higher-dimensional simplicial complexes). Studies have shown that higher-order structures play a crucial role in various scenarios, such as group relationships in social networks (Alvarez-Rodriguez et al., 2021; Bick et al., 2023; Boccaletti et al., 2023) and atomic configurations in molecular networks (Morris et al., 2019; Doye & Massen, 2005). However, current methods generally lack effective modeling of these higher-order structures, leading to suboptimal performance when capturing complex graph interactions, especially in graphs dominated by higher-order structures. Furthermore, the modeling of low-order relationships limits the scope and depth of information propagation, failing to capture long-range dependencies in graphs, whereas higher-order structures can better bridge local and global features. The absence of higher-order structure modeling not only hampers global information transmission but also restricts the recognition of complex graph patterns, particularly in large-scale sparse graphs (Jin et al., 2022).

(2) Global diffusion leads to the loss of local information: While the introduction of global diffusion mechanisms has effectively captured large-scale global structural information and mitigated the over-smoothing problem, the global diffusion process often overlooks local detail information and comes at the expense of sacrificing local patterns (Zhao et al., 2021; Chamberlain et al., 2021; Thorpe et al., 2022; Gasteiger et al., 2019; Li et al., 2024; Liu et al., 2024). In large-scale sparse graphs or graphs with pronounced community structures, global diffusion may result in over-propagation of information between nodes, diluting or losing crucial local information (Long et al., 2020; Li et al., 2022). For instance, in social networks, local interactions between users often exhibit highly personalized and fine-grained characteristics, which are critical for tasks such as recommendation systems or influence propagation (Song et al., 2019; Wilson et al., 2009). However, the global diffusion approach in existing models may cause these fine-grained local patterns to be diluted under the influence of global information, leading to an inability of the model to fully perceive and utilize these critical pieces of information. Therefore, an important research challenge is how to enhance sensitivity to local structures while simultaneously conducting global diffusion.

(3) Absence of a unified diffusion mechanism for multi-level structures: Despite recent efforts to incorporate higher-order structures (such as simplicial complexes and hypergraphs) into information propagation frameworks (Prokopchik et al., 2022; Liu et al., 2021), most existing methods treat the diffusion mechanisms for nodes, edges, and higher-order structures independently, lacking a unified approach to handle multi-level structures. Complex interactions often exist between different levels of structures, and current graph diffusion methods tend to exhibit imbalance and fragmentation when handling these interactions, resulting in ineffective transmission of information across different structural levels. Although the introduction of simplicial complexes provides a way to model higher-order structures (Yang et al., 2022; Chen et al., 2022; Benson et al., 2018), how to unify the propagation of information across nodes, edges, and higher-order structures remains a largely unsolved problem.

In this paper, we propose a generalized hierarchical diffusion framework based on simplicial complexes (HDSC) to address the limitations of existing diffusion models and graph neural networks (GNNs) in handling higher-order structures. Specifically, HDSC defines hierarchical diffusion using simplicial complexes of different dimensions and introduces boundary operators to connect higher-order geometric structures with lower-order ones, ensuring efficient information propagation across various levels. To maintain topological consistency during information transmission, we employ a high-order Laplacian operator to guide the hierarchical diffusion and ensure stability in the diffusion

process through the asymptotic decay of eigenvalues. Additionally, we design a time-dependent enhanced topological memory mechanism that strengthens the model's structural awareness during diffusion, preventing rapid information loss during learning.

The main contributions of this paper are threefold:

- We present a unified framework for multi-level diffusion through simplicial complexes, capable of handling feature diffusion across nodes, edges, and higher-order geometric structures simultaneously. This allows for efficient coupling of information across different levels, enhancing both the global stability and local feature-capturing ability of the model.

- We design a time-dependent enhanced topological memory mechanism that preserves historical information from different topological levels during the learning process. By capturing local dynamic changes, it ensures the consistency and coherence of information, thereby improving the model's capacity to handle long-range dependencies.

- Extensive experiments demonstrate that our proposed HDSC framework outperforms existing methods on multiple benchmark datasets, validating the effectiveness of multi-level diffusion in propagating higher-order geometric structure information.

## 2 PROBLEM SETUP

In this section, we formally introduce the problem setting for generalized hierarchical diffusion and elaborate on the main assumptions adopted in this work. Except where stated otherwise, we will focus on the following setting:

- **Higher-order structures in graphs**: Consider a graph $G = (V, E, \mathbf{X})$, where the node set $V = \{v_1, v_2, \ldots, v_n\}$ and edge set the edge set $E \subseteq \{(v_i, v_j) \mid v_i, v_j \in V, i \neq j\}$ describes the connections between nodes as unordered pairs, $\mathbf{X} = \{\mathbf{x}_1, \mathbf{x}_2, \ldots, \mathbf{x}_n\}$ is the set of node feature vectors, where each $\mathbf{x}_i \in \mathbb{R}^d$ represents the feature of node $\mathbf{v}_i$. To capture higher-order geometric information in the graph, we introduce the $k$-simplicial complexes $S_k \subseteq \text{Select}(V, k+1)$, where $\text{Select}(V, k+1)$ denotes the set of all combinations of $k+1$ nodes from $V$. A $k$-simplicial complexes is formed by $k + 1$ nodes that are all mutually connected, meaning that every subset of these $k + 1$ nodes forms a lower-order simplicial complexes. For example, a 2-simplicial complexes (triangles) consists of 3 nodes, where each pair of nodes is connected by an edge. The information representation of node $v_i$ at level $k$ is denoted as $\mathbf{x}_i^{(k)} \in \mathbb{R}^d$, i.e., $\mathbf{x}_i^{(k)}$ is the feature vector of node $v_i$ within the $k$-simplicial complexes. These representations participate in information propagation and exchange across different levels of higher-order structures.

- **Hierarchical diffusion process**: The process of diffusion on a graph can be understood as the propagation of features (such as signals or information) between nodes through the graph structure. In the context of graphs, the diffusion equation can be described as capturing the difference between the states of a node and its neighboring nodes:

$$\frac{\mathrm{d}\mathbf{X}^{(t)}}{\mathrm{d}t} = -\mathbf{L}\mathbf{X}^{(t)}, \tag{1}$$

where $\mathbf{X}^{(t)} \in \mathbb{R}^{n \times d}$ is the state matrix of the nodes at time $t$, and $\mathbf{L} \in \mathbb{R}^{n \times n}$ is the graph Laplacian matrix defined as $\mathbf{L} = \mathbf{D} - \mathbf{A}$, with $\mathbf{A} \in \mathbb{R}^{n \times n}$ being the adjacency matrix of the graph and $\mathbf{D} \in \mathbb{R}^{n \times n}$ is the degree matrix, where each diagonal entry $D_{ii}$ represents the degree of node $v_i$, calculated as the sum of the weights of all edges connected to $v_i$. By discretizing the time variable with a small time step $\Delta t$ and applying Euler's forward method to approximate the continuous time derivative, we obtain

$$\mathbf{X}^{(t+1)} = \mathbf{X}^{(t)} - \Delta t \mathbf{L} \mathbf{X}^{(t)}. \tag{2}$$

Equation (2) governs how node features propagate through the graph topology at each time step $t$. For the hierarchical diffusion process, we use $\mathbf{L}_k$ to denote the $k$-th-level laplacian operator that acts on the $k$-simplicial structure. The corresponding diffusion behavior across different levels can be formulated as follows:

$$\mathbf{X}^{(t+1,k)} = \mathbf{X}^{(t,k)} - \Delta t \mathbf{L}_k \mathbf{X}^{(t,k)}, \tag{3}$$

where $\mathbf{X}^{(t,k)}$ represents the state matrix of the nodes at time $t$ and level $k$.

- **Topological consistency**: In hierarchical diffusion process, the propagation of information across nodes, edges, and higher-order structures need to be topologically consistent. Specifically, we assume the information transfer follows the boundary operator relationship:

$$\mathbf{B}_k \circ \mathbf{B}_{k-1} = 0, \tag{4}$$

where $\mathbf{B}_k$ is the boundary operator for the $k$-simplicial complex, defined as $\mathbf{B}_k \in \mathbb{R}^{n \times n}$, which transfers information from higher-order to lower-order structures, ensuring that information is propagated across different levels without introducing redundancy or distortion. Here, the symbol "$\circ$" represents the composition of operators, indicating the sequential application of two boundary operators $\mathbf{B}_k$ and $\mathbf{B}_{k-1}$. To further ensure consistency, we introduce the following normalized boundary operator

$$\tilde{\mathbf{B}}_k = \mathbf{D}_k^{-1/2} \mathbf{B}_k, \tag{5}$$

where $\mathbf{D}_k \in \mathbb{R}^{n \times n}$ is the degree matrix of the $k$-simplicial complexes, representing the inverse square root of the degree associated with each node.

# 3 METHODOLOGY

This section provides a detailed description of our proposed hierarchical diffusion model and its associated mechanisms, aiming to achieve adaptive diffusion from local to global levels by introducing multi-order simplicial complexes structures (such as nodes, edges, and triangles). To accomplish this, we construct a unified generalized hierarchical diffusion equation, combined with a topological memory mechanism and a generalized energy optimization strategy, ensuring efficient information propagation and convergence within complex graph structures.

## 3.1 HIERARCHICAL DIFFUSION EQUATION

In this section, we begin with the basic diffusion equation and gradually extend the diffusion process to 1-simplicial complexes and 2-simplicial complexes, ultimately deriving a unified generalized hierarchical diffusion equation.

To convert the discrete diffusion mechanism into a continuous-time representation, for Equation (3), we consider the limiting case where the time step $\Delta t \rightarrow 0$. In this limit, the difference form is transformed into a derivative, yielding the continuous-time fundamental diffusion equation:

$$\frac{\mathrm{d}\mathbf{X}^{(t,k)}}{\mathrm{d}t} = -\mathbf{L}_k \mathbf{X}^{(t,k)}, \quad \mathbf{X}^{(0,0)} = \mathbf{X}, \tag{6}$$

Thus, for the 0-simplicial complexes (nodes level), the feature evolution of node $i$ can be described through the feature changes of its neighboring nodes $j$:

$$\frac{\mathrm{d}\mathbf{x}_i^{(t,0)}}{\mathrm{d}t} = -\sum_{j \in N(i)} \mathbf{A}_{ij} \left( \mathbf{x}_i^{(t,0)} - \mathbf{x}_j^{(t,0)} \right), \tag{7}$$

where $\mathbf{x}_i^{(t,0)}$ represents the feature of node $i$ at time $t$, $\mathbf{A}_{ij} \in \mathbb{R}^{n \times n}$ denotes the connectivity between nodes $i$ and $j$, and $N(i)$ is the set of neighbors of node $i$. The negative sign indicates that the features tend to homogenize during the diffusion process, i.e., node features converge towards their neighboring node features. Building upon the diffusion in 0-simplicial complexes, the diffusion mechanism in 1-simplicial complexes (edges level) is governed by the boundary operator $\mathbf{B}_1$:

$$\mathbf{X}^{(t+1,1)} = \mathbf{X}^{(t,1)} - \Delta t \mathbf{L}_1 \mathbf{X}^{(t,1)}, \quad \mathbf{L}_1 = \mathbf{B}_1^T \mathbf{B}_1, \tag{8}$$

where $\mathbf{X}^{(t,1)}$ represents the features of the 1-simplicial complexes at time $t$, $\mathbf{B}_1 \in \mathbb{R}^{|E| \times n}$ maps node features to edge features, and $\mathbf{L}_1 \in \mathbb{R}^{|E| \times |E|}$ is the laplacian operator acting on edges, describing the evolution of edge features during diffusion. For each edge $e_{ij}$, we have:

$$\frac{\mathrm{d}\mathbf{x}_{e_{ij}}^{(t,1)}}{\mathrm{d}t} = -\sum_{\tau \in N_2(e_{ij})} \mathbf{B}_1^T \left( \mathbf{x}_{e_{ij}}^{(t,1)} - \mathbf{x}_\tau^{(t,1)} \right), \quad \mathbf{x}_{e_{ij}}^{(0,1)} = \mathbf{B}_1(\mathbf{x}_i^{(0,0)} - \mathbf{x}_j^{(0,0)}), \tag{9}$$

where $\mathbf{x}_{e_{ij}}^{(t,1)}$ represents the feature of edge $e_{ij}$ at time $t$, $N_2(e_{ij})$ is the set of neighboring triangles of edge $e_{ij}$. Similarly, for the diffusion mechanism in 2-simplicial complexes (triangles level), we introduce the boundary operator $\mathbf{B}_2$, which maps edge features to the triangle feature space:

$$\mathbf{X}^{(t+1,2)} = \mathbf{X}^{(t,2)} - \Delta t \mathbf{L}_2 \mathbf{X}^{(t,2)}, \quad \mathbf{L}_2 = \mathbf{B}^T \mathbf{B}_2, \tag{10}$$

where $\mathbf{X}^{(t,2)} \in \mathbb{R}^{|T| \times d}$ represents the features of triangles at time $t$, $\mathbf{B}_2 \in \mathbb{R}^{|T| \times |E|}$ maps edges to triangles, and $|T|$ denotes the number of triangles. The Laplacian operator $\mathbf{L}_2 \in \mathbb{R}^{|T| \times |T|}$ acts on 2-simplicial complexes, describing feature diffusion based on the adjacency relations between triangles. For instance, if two triangles share an edge, the difference in their features influences the rate of diffusion between them. Therefore, the diffusion equation for 2-simplicial complexes is extended as:

$$\frac{\mathrm{d}\mathbf{x}_\sigma^{(t,2)}}{\mathrm{d}t} = - \sum_{\pi \in N_3(\sigma)} \mathbf{B}_2^T \left( \mathbf{x}_\sigma^{(t,2)} - \mathbf{x}_\pi^{(t,2)} \right), \quad \mathbf{x}_\sigma^{(0,2)} = \mathbf{B}_2(\mathbf{x}_{e_{ij}}^{(0,1)} - \mathbf{x}_\tau^{(0,1)}), \tag{11}$$

where $\mathbf{x}_\sigma^{(t,2)}$ represents the feature of the 2-simplicial complexes, $N_3(\sigma)$ refers to other triangles that share an edge with the triangle $\sigma$. Finally, we obtain the unified expression for generalized hierarchical diffusion:

$$\mathbf{X}^{(t+1)} = \mathbf{X}^{(t)} - \Delta t \sum_{k=0}^{K} \theta_{(t,k)} \tilde{\mathbf{B}}_k^T \tilde{\mathbf{B}}_k \mathbf{X}^{(t,k)}, \quad \tilde{\mathbf{B}}_k \in \mathbb{R}^{|S_k| \times |S_{k-1}|}, \tag{12}$$

$$\frac{\mathrm{d}\mathbf{X}^{(t)}}{\mathrm{d}t} = - \sum_{k=0}^{K} \theta_{(t,k)} \tilde{\mathbf{B}}_k^T \tilde{\mathbf{B}}_k \mathbf{X}^{(t,k)}, \tag{13}$$

where $\mathbf{X}^{(t+1)}$ represents the node features at time $t+1$, which integrate multi-level diffusion information. $\theta_{(t,k)}$ is a time-dependent diffusion coefficient controlling the diffusion rate at level $k$ and time $t$, $\tilde{\mathbf{B}}_k$ is the normalized boundary operator for $k$-simplicial complexes, and $\mathbf{L}_k \in \mathbb{R}^{|S_k| \times |S_k|}$ is the higher-order Laplacian operator acting on $k$-simplicial complexes. Equations (12) and (13) describe how information propagates cooperatively across the multi-level structure of the graph.

## 3.2 Time-Dependent Enhanced Topological Memory Mechanism

Different levels of simplicial complexes are often treated as independent entities in the propagation process, leading to a disconnection between the information in higher-order structures and lower-order structures. This results in two major issues:

- **Information Fragmentation**: Local topological features in lower-order structures cannot be effectively transferred to higher-order structures, thus limiting the propagation of global topological information.

- **Lack of Feedback from Higher-Order Structures**: The overall features of higher-order structures cannot be reflected back to lower-order structures (such as nodes and edges), preventing the adjustment of low-dimensional feature propagation paths.

Based on the above, this paper designs a time-dependent enhanced topological memory mechanism that captures historical topological information of simplicial complexes across different dimensions through memory units. Specifically, at each time step $k$, a gating mechanism is used to update the memory units to adapt to current local changes while retaining historical topological features. This mechanism primarily consists of two components:

- **Update Gate**: Controls the weighted update between the current features and historical memory:

$$\mathbf{u}^{(t,k)} = \sigma(\mathbf{W}_u^{(t,k)}[\mathbf{X}^{(t,k)}, \Phi^{(t,k)}] + \mathbf{b}_u^{(t,k)}), \tag{14}$$

where $\mathbf{W}_u^{(t,k)}$ is the learnable weight parameter matrix, $\mathbf{b}_u^{(t,k)}$ is the bias. $\mathbf{X}^{(t,k)}$ represents the features of the $k$-simplicial complexes at time step $t$, and $\Phi^{(t,k)}$ represents the historical memory of the same simplicial complex from previous time steps.

- **Forget Gate**: Controls the extent to which historical information is forgotten:

$$\mathbf{r}^{(t,k)} = \sigma(\mathbf{W}_r^{(t,k)}[\mathbf{X}^{(t,k)}, \Phi^{(t,k)}] + \mathbf{b}_r^{(t,k)}), \tag{15}$$

where $\mathbf{W}_r^{(t,k)}$ is the learnable weight parameter matrix, $\mathbf{b}_r^{(t,k)}$ is the bias. The forget gate $\mathbf{r}_t$ regulates the impact of the historical memory $\Phi^{(t,k)}$ on the current time step $t$, allowing the system to retain long-term topological memory while adapting to short-term local changes.

At time step $t+1$, the historical memory is updated based on the current features $\mathbf{X}^{(t,k)}$ and the historical memory $\Phi^{(t,k)}$. Taking the 0-simplicial complexes historical memory as an example, the update process is as follows:

$$\tilde{\Phi}_{\text{node}}^{(t+1,0)} = \tanh(\mathbf{W}_{\tilde{\Phi}}^{(t,0)}[\mathbf{X}^{(t,0)}, \mathbf{r}_t \odot \Phi_{\text{node}}^{(t,0)}] + \mathbf{b}_{\tilde{\Phi}}^{(t,0)}), \tag{16}$$

$$\Phi_{\text{node}}^{(t+1,0)} = \left(1 - \mathbf{u}^{(t,0)}\right) \odot \Phi_{\text{node}}^{(t,0)} + \mathbf{u}^{(t,0)} \odot \tilde{\Phi}_{\text{node}}^{(t+1,0)}, \tag{17}$$

where $\odot$ represents the Hadamard product, $\tilde{\Phi}_{\text{node}}^{(t+1,0)}$ represents the candidate memory state adjusted by the forget gate $\mathbf{r}_t$, and $\Phi_{\text{node}}^{(t+1,0)}$ represents the final memory state obtained through the weighted fusion by the update gate $\mathbf{u}^{(t,0)}$. Incorporating the time-dependent enhanced topological memory mechanism, the final generalized hierarchical diffusion equation can be expressed as:

$$\mathbf{X}^{(t+1)} = \mathbf{X}^{(t)} - \Delta t \sum_{k=0}^{K} \theta_{(t,k)} \tilde{\mathbf{B}}_k^T \tilde{\mathbf{B}}_k \mathbf{X}^{(t,k)} + \alpha \Omega^{(t)}, \tag{18}$$

where $\alpha$ denotes the memory decay coefficient, controlling the influence of long-term feature changes, and $\Omega^{(t)}$ represents the historical information retrieved from the topological memory units at time $t$, defined as:

$$\Omega^{(t)} = \zeta_{\text{node}} \Phi_{\text{node}}^{(t,0)} + \zeta_{\text{edge}} \Phi_{\text{edge}}^{(t,1)} + \zeta_{\text{triangle}} \Phi_{\text{triangle}}^{(t,2)}, \tag{19}$$

where $\zeta_{\text{node}}, \zeta_{\text{edge}}, \zeta_{\text{triangle}}$ control the contributions of the historical features from the node, edge, and triangle memory units, respectively, to the current diffusion process.

---

**Algorithm 1** HDSC

---

**Input:** Graph data $\mathcal{G} = (V, E, \mathbf{X})$, diffusion steps $T$, memory steps $m$,
**Output:** Total loss $\mathcal{L}$
1: **repeat**
2:     **for** Epoch = 1, 2, ..., MaxEpoch **do**
3:         **for** Step $t = 1$ to $T$ **do**
4:             Compute hierarchical diffusion features $\mathbf{X}^{(t+1)}$ from Equation (13).
5:             **if** $t\%m == 0$ **then**
6:                 Update memory units parameters $\Phi_{\text{nmu}}$ , $\Phi_{\text{emu}}$, $\Phi_{\text{tmu}}$ from Equation (17).
7:                 Compute memory effect $\Omega$ from Equation (19).
8:             **end if**
9:             Update node features $\mathbf{X}^{(t+1)}$ from Equation (18).
10:         **end for**
11:         Use the Adam optimizer to update parameters to minimize the total loss $\mathcal{L}$
12:     **end for**
13: **until** Convergence of the total loss $\mathcal{L}$

---

The specific algorithm flow is shown in Algorithm 1. At each step, the node features are updated based on the hierarchical diffusion equation. Then, at specified memory intervals, the memory units for nodes, edges, and triangles are updated, with the corresponding historical information stored to enhance the long-term propagation of information. Let the total loss $\mathcal{L}$ be as follows:

$$\mathcal{L} = -\sum_{i=1}^{n} y_i \log(\hat{y}_i), \tag{20}$$

where $y_i$ is the truth label, $\hat{y}_i$ is the predicted probability. Finally, by adaptively adjusting the diffusion rate and memory effects, the process is optimized across multiple levels of the structure until the total loss function $\mathcal{L}$ converges, ensuring efficient information dissemination.

## 4 EXPERIMENT

**Dataset.** For a comprehensive evaluation of the proposed HDSC model, we conduct experiments on six widely used benchmark datasets that represent various real-world graph structures. These datasets include three citation networks, namely Cora, Citeseer, Pubmed (Yang et al., 2016), one co-authorship network CoauthorCS (Shchur et al., 2018), and two co-purchase networks, namely Computer and Photo (Shchur et al., 2018). Details of the datasets can be found in Appendix.

| Number of samples per class | Model | Cora | Citeseer | Pubmed |
|---|---|---|---|---|
| 1 | MoNet | 47.72 ± 15.53 | 39.13 ± 11.37 | 56.47 ± 4.67 |
| | GCN | 47.72 ± 15.33 | 48.94 ± 10.24 | 58.61 ± 12.83 |
| | GAT | 47.86 ± 15.38 | 50.31 ± 14.27 | 58.84 ± 12.81 |
| | GraphSage | 43.04 ± 14.01 | 48.81 ± 11.45 | 55.53 ± 12.71 |
| | GRAND | 52.53 ± 16.40 | 50.06 ± 17.98 | 62.11 ± 10.58 |
| | GRAND++ | 54.94 ± 16.09 | **58.95** ± 9.59 | **65.94** ± 4.87 |
| | HDSC | **65.63** ± 7.59 | 56.28 ± 6.79 | 62.06 ± 5.66 |
| 2 | MoNet | 60.85 ± 14.01 | 48.52 ± 9.52 | 61.03 ± 6.93 |
| | GCN | 60.85 ± 14.01 | 58.06 ± 9.76 | 60.45 ± 16.20 |
| | GAT | 58.30 ± 13.55 | 55.55 ± 9.19 | 60.24 ± 14.44 |
| | GraphSage | 53.96 ± 12.18 | 54.39 ± 11.37 | 58.97 ± 12.65 |
| | GRAND | 64.82 ± 11.16 | 59.55 ± 10.89 | 69.00 ± 7.55 |
| | GRAND++ | 66.92 ± 10.04 | 64.98 ± 8.31 | 69.31 ± 4.87 |
| | HDSC | **78.92** ± 1.55 | **65.69** ± 4.13 | **70.28** ± 5.02 |
| 5 | MoNet | 73.86 ± 7.97 | 61.66 ± 6.61 | 67.92 ± 2.50 |
| | GCN | 73.86 ± 7.97 | 67.24 ± 4.19 | 68.69 ± 7.93 |
| | GAT | 71.04 ± 5.74 | 67.37 ± 5.08 | 68.54 ± 5.75 |
| | GraphSage | 68.14 ± 6.95 | 64.79 ± 5.16 | 66.07 ± 6.16 |
| | GRAND | 76.07 ± 5.08 | 68.37 ± 5.00 | 73.98 ± 5.08 |
| | GRAND++ | 77.80 ± 4.46 | 70.03 ± 3.63 | 71.99 ± 1.91 |
| | HDSC | **82.50** ± 0.85 | **72.31** ± 1.16 | **75.80** ± 3.82 |
| 10 | MoNet | 78.82 ± 5.38 | 68.08 ± 6.29 | 71.24 ± 1.54 |
| | GCN | 78.82 ± 5.38 | 72.18 ± 3.47 | 72.59 ± 3.19 |
| | GAT | 76.31 ± 4.87 | 71.35 ± 4.92 | 72.44 ± 3.50 |
| | GraphSage | 75.04 ± 5.03 | 68.90 ± 5.08 | 70.74 ± 3.11 |
| | GRAND | 80.25 ± 3.40 | 71.90 ± 7.66 | 76.33 ± 3.41 |
| | GRAND++ | 80.86 ± 2.99 | 72.34 ± 2.42 | 75.13 ± 3.88 |
| | HDSC | **84.34** ± 0.77 | **73.81** ± 0.98 | **82.49** ± 1.04 |
| 20 | MoNet | 82.07 ± 2.03 | 71.52 ± 4.11 | 76.49 ± 1.75 |
| | GCN | 82.07 ± 2.03 | 74.21 ± 2.90 | 76.89 ± 3.27 |
| | GAT | 79.92 ± 2.28 | 73.22 ± 2.90 | 75.55 ± 4.11 |
| | GraphSage | 80.04 ± 2.54 | 72.02 ± 2.82 | 74.55 ± 3.09 |
| | GRAND | 80.25 ± 3.40 | 71.90 ± 7.66 | 76.33 ± 3.41 |
| | GRAND++ | 82.95 ± 1.37 | 73.53 ± 3.31 | 79.16 ± 1.37 |
| | HDSC | **85.75** ± 0.91 | **74.63** ± 1.39 | **84.11** ± 0.92 |

Table 1: Performance of different datasets. (datasets: Cora, Citeseer, Pubmed)

**Baselines.** The proposed HDSC is compared with four conventional graph neural network models, including GCN (Kipf & Welling, 2017), GraphSAGE (Hamilton et al., 2017), GAT (Veličković et al., 2018) and MoNet (Monti et al., 2017). Additionally, it is compared with two graph diffusion-based models, GRAND (Chamberlain et al., 2021) and GRAND++ (Thorpe et al., 2022), to assess the performance of different models on diverse graph structures.

**Results.** Tables 1 and 2 summarize the test results in terms of accuracy for the node classification task. It can be observed that for most cases, HDSC significantly outperforms other baseline models across six datasets. HDSC effectively integrates geometric information from different dimensions

| Number of samples per class | Model | CoauthorCS | Computer | Photo |
|---|---|---|---|---|
| 1 | MoNet | 58.99 ± 5.17 | 23.78 ± 7.57 | 34.72 ± 8.18 |
| | GCN | 65.22 ± 2.25 | 49.46 ± 1.65 | 82.94 ± 2.17 |
| | GAT | 51.13 ± 5.24 | 37.14 ± 7.81 | 73.58 ± 8.15 |
| | GraphSage | 61.35 ± 1.35 | 27.65 ± 2.39 | 45.36 ± 7.13 |
| | GRAND | 59.15 ± 5.73 | 48.67 ± 1.66 | 81.25 ± 2.50 |
| | GRAND++ | 60.30 ± 1.50 | **67.65** ± 0.37 | 83.12 ± 0.78 |
| | HDSC | **61.02** ± 0.78 | 65.87 ± 2.12 | **83.68** ± 1.38 |
| 2 | MoNet | 76.57 ± 4.06 | 38.19 ± 3.72 | 43.03 ± 8.22 |
| | GCN | **83.61** ± 1.49 | 76.90 ± 1.49 | 83.61 ± 0.71 |
| | GAT | 63.12 ± 6.09 | 65.07 ± 8.86 | 76.89 ± 4.89 |
| | GraphSage | 76.51 ± 1.31 | 42.63 ± 4.29 | 51.93 ± 4.21 |
| | GRAND | 73.83 ± 5.58 | 74.77 ± 1.85 | 82.13 ± 3.27 |
| | GRAND++ | 76.53 ± 1.85 | **76.47** ± 1.48 | 83.71 ± 0.90 |
| | HDSC | 77.33 ± 1.32 | 75.67 ± 0.56 | **84.14** ± 0.45 |
| 5 | MoNet | 87.02 ± 1.67 | 59.38 ± 4.73 | 71.80 ± 5.02 |
| | GCN | 86.66 ± 0.43 | 82.47 ± 0.97 | 88.86 ± 1.56 |
| | GAT | 71.65 ± 4.53 | 71.43 ± 7.34 | 83.01 ± 3.64 |
| | GraphSage | **89.06** ± 0.69 | 64.83 ± 1.62 | 78.26 ± 1.93 |
| | GRAND | 85.29 ± 2.19 | 80.72 ± 1.09 | 88.27 ± 1.94 |
| | GRAND++ | 84.83 ± 0.84 | 82.64 ± 0.56 | 88.33 ± 1.21 |
| | HDSC | 85.35 ± 1.13 | **82.78** ± 0.55 | **89.57** ± 0.77 |
| 10 | MoNet | 88.76 ± 0.49 | 68.66 ± 3.30 | 78.66 ± 3.17 |
| | GCN | 88.60 ± 0.50 | 82.53 ± 0.74 | 90.41 ± 0.35 |
| | GAT | 74.71 ± 3.35 | 76.04 ± 0.35 | 87.42 ± 2.38 |
| | GraphSage | **89.68** ± 0.39 | 74.66 ± 1.29 | 84.38 ± 1.75 |
| | GRAND | 87.81 ± 1.36 | 82.42 ± 1.10 | 90.98 ± 0.93 |
| | GRAND++ | 86.94 ± 0.46 | 82.99 ± 0.81 | 90.65 ± 1.19 |
| | HDSC | 85.85 ± 0.67 | **83.75** ± 0.43 | **91.12** ± 0.35 |
| 20 | MoNet | 90.31 ± 0.41 | 73.66 ± 2.87 | 88.61 ± 1.18 |
| | GCN | 91.09 ± 0.35 | 82.94 ± 1.54 | 91.95 ± 0.11 |
| | GAT | 79.95 ± 2.88 | 80.05 ± 1.81 | 89.38 ± 2.48 |
| | GraphSage | **91.33** ± 0.36 | 79.98 ± 0.96 | 91.29 ± 0.67 |
| | GRAND | 91.03 ± 0.47 | 84.54 ± 0.90 | 93.53 ± 0.47 |
| | GRAND++ | 90.80 ± 0.34 | 85.73 ± 0.50 | 93.55 ± 0.38 |
| | HDSC | **91.33** ± 0.45 | **86.48** ± 0.47 | **94.54** ± 0.31 |

Table 2: Performance of different datasets. (datasets: CoauthorCS, Computer, Photo)

from a hierarchical perspective, ensuring cooperative diffusion and propagation between features at various levels. Additionally, the introduction of the topological memory mechanism allows the model to efficiently retain historical information and optimize the information transmission paths.

**Visualization of Clustering Results.** Figure 1 illustrates the clustering results, demonstrating that HDSC significantly outperforms other baseline models. Specifically, HDSC displays more distinct clustering patterns with clearer boundaries between different categories. This indicates that the hierarchical feature representation of HDSC effectively integrates information across multiple levels, thereby reducing the degree of information mixing. These findings underscore the superiority of HDSC in both information diffusion and structural capture.

**Impact of $\alpha$.** We tested the impact of $\alpha$ in Equation (18), and as shown in Figure 2, for lower values of $\alpha$, the model relies more on current node or local structural information. This effect is particularly pronounced when the number of sample labels is small, as the influence of global topological information has not yet fully emerged. However, as $\alpha$ increases, more historical topological information is gradually integrated, improving overall performance on complex datasets. This is

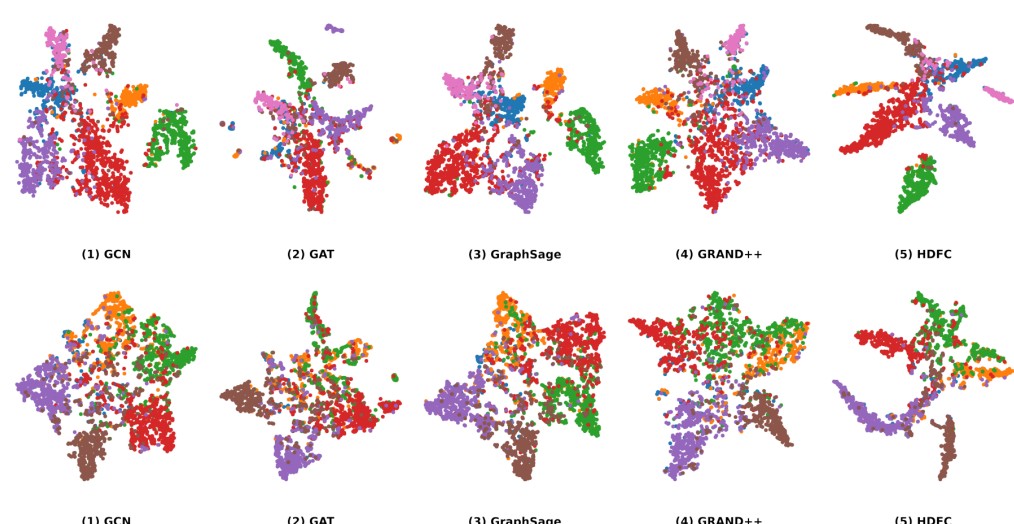

Figure 1: Visualization of clustering results. (The first row illustrates the performance on the Cora dataset, and the second row illustrates the performance on the Citeseer dataset.)

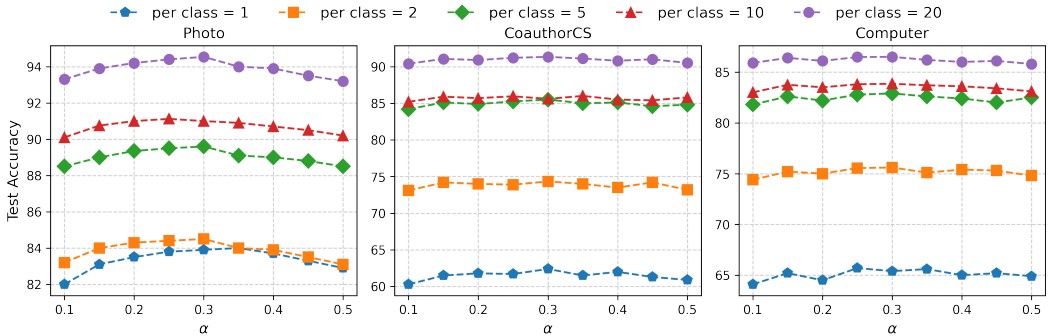

Figure 2: The impact of different $\alpha$ values on performance.

especially evident when the number of labels is larger, where the introduction of global topological memory effectively captures cross-level information interactions, thereby enhancing the model's performance. Nevertheless, when $\alpha$ becomes too large, the model may over-rely on global information, weakening the contribution of local geometric features.

## 5 CONCLUSION AND FUTURE WORK

This paper presents a novel framework, HDSC, which enables adaptive information diffusion across various levels of simplicial complexes and achieves efficient coupling between different structural levels. Additionally, by incorporating a time-dependent topological memory mechanism, the framework significantly enhances the smoothness and coherence of global information flow, allowing multi-layer features to diffuse collaboratively. Experimental results demonstrate that HDSC exhibits outstanding performance in downstream tasks and effectively avoids the phenomenon of node feature homogenization. A potential direction for future work could be to explore ways of integrating more complex topological features and contextual information into the model, thereby enhancing its capacity to handle non-stationary data.

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

## A  REPRODUCIBILITY INFORMATION

**Dataset.** Table 3 provides detailed information about each dataset.

| Dataset | Nodes | Edges | Features | Classes |
|---|---|---|---|---|
| Cora (Yang et al., 2016) | 2,708 | 5,429 | 1,433 | 7 |
| Citeseer (Yang et al., 2016) | 3,327 | 4,732 | 3,703 | 6 |
| Pubmed (Yang et al., 2016) | 19,717 | 44,338 | 500 | 3 |
| CoauthorCS (Shchur et al., 2018) | 18,333 | 81,894 | 6,805 | 15 |
| Amazon-Computer (Shchur et al., 2018) | 13,752 | 574,418 | 767 | 10 |
| Amazon-Photo (Shchur et al., 2018) | 7,650 | 238,162 | 745 | 8 |

Table 3: Datasets Overview

**Experimental Setup.** All experiments were conducted on a machine equipped with an NVIDIA L40 GPU, an Intel Core i9 processor, and 128GB of RAM. The experimental code is based on Python 3.8, PyTorch 1.12.1, and the PyTorch Geometric (PyG) library for graph data processing. To evaluate the performance of the model for varying number of samples, we vary the number of samples per class within $\{1, 2, 5, 10, 20\}$. For all methods, we performed 10 random trials and reported the mean and variance over these 10 random trials. It is worth noting that, following the same principle, the datasets Amazon-Computer and Amazon-Photo were processed using their largest connected subgraphs.

| Dataset | $\alpha$ | $\zeta_{\text{node}}$ | $\zeta_{\text{edge}}$ | $\zeta_{\text{triangle}}$ | lr | weight decay | hidden units | dropout |
|---|---|---|---|---|---|---|---|---|
| Cora | 0.3 | 1 | 0.15 | 0.01 | 0.01 | 0.08 | 64 | 0.5 |
| Citeseer | 0.3 | 1 | 0.15 | 0.03 | 0.01 | 10.2 | 64 | 0.2 |
| Pubmed | 0.3 | 1.0 | 0.1 | 0.02 | 0.01 | 0.03 | 64 | 0.5 |
| CoauthorCS | 0.25 | 0.9 | 0.15 | 0.02 | 0.01 | 0.06 | 64 | 0.5 |
| Computer | 0.2 | 0.95 | 0.2 | 0.02 | 0.01 | 0.03 | 64 | 0.5 |
| Photo | 0.25 | 0.9 | 0.15 | 0.02 | 0.01 | 0.03 | 64 | 0.5 |

Table 4: Hyperparameters for Different Datasets

The optimal parameters of HDSC for each dataset are shown in Table 4. Specifically, $\alpha$ is the coefficient controlling the influence of memory. $\zeta_{\text{node}}$, $\zeta_{\text{edge}}$, and $\zeta_{\text{triangle}}$ represent the memory coefficients for nodes, edges, and triangles, respectively, reflecting the impact of historical information from different topological structures on the model. lr is the learning rate, which determines the step size for updating model parameters. weight_decay is the weight decay coefficient, used to prevent overfitting in the model. hidden_units refers to the number of units in the hidden layer, affecting the model's capacity. dropout is the dropout rate used to prevent overfitting, controlling the proportion of neurons randomly ignored during each training session.

**Other Source Code.** The acquisition of all the code below complies with the provider's license and do not contain personally identifiable information and offensive content. The link to the code of baselines are listed as follows:

- GCN (MIT license): https://github.com/tkipf/gcn

- GAT (MIT license): https://github.com/gordicaleksa/pytorch-GAT

- GraphSage (MIT license): https://github.com/williamleif/GraphSAGE

- MoNet (MIT license) : https://github.com/sw-gong/MoNet

- GRAND (MIT license) : https://github.com/twitter-research/graph-neural-pde

- GRAND++ (MIT license) : https://github.com/twitter-research/graph-neural-pde

# B  MORE RESULTS

**Ablation Study.** Tables 5, 6, 7, 8, and 9 display the ablation experiments of HDSC under different numbers of samples per class. HDSC-1 relies solely on the diffusion mechanism of 0-simplicial complexes; HDSC-2 incorporates 1-simplicial complexes based on HDSC-1; and HDSC employs a multi-level diffusion mechanism utilizing 0-simplicial complexes, 1-simplicial complexes, and 2-simplicial complexes.

| Model | Cora | Citeseer | Pubmed | CoauthorCS | Computer | Photo |
|---|---|---|---|---|---|---|
| HDSC-1 | 56.79 ± 7.15 | 55.65 ± 6.12 | 58.89 ± 4.15 | 58.45 ± 0.85 | 66.35 ± 1.44 | 81.44 ± 0.59 |
| HDSC-2 | 60.35 ± 5.45 | **56.46** ± 5.75 | 61.79 ± 4.34 | 59.37 ± 1.48 | **67.04** ± 1.08 | 82.36 ± 1.64 |
| HDSC | **65.63** ± 7.59 | 56.28 ± 6.79 | **62.06** ± 5.66 | **61.02** ± 0.78 | 65.87 ± 2.12 | **83.68** ± 1.38 |

Table 5: Ablation study results. (The number of samples per class is 1)

| Model | Cora | Citeseer | Pubmed | CoauthorCS | Computer | Photo |
|---|---|---|---|---|---|---|
| HDSC-1 | 67.44 ± 2.41 | 64.32 ± 3.56 | 69.04 ± 4.30 | 73.23 ± 0.87 | 74.65 ± 0.89 | 82.78 ± 0.85 |
| HDSC-2 | 74.48 ± 2.38 | 65.01 ± 2.11 | 69.66 ± 3.75 | **74.87** ± 0.67 | 75.03 ± 0.62 | 83.98 ± 0.53 |
| HDSC | **78.92** ± 1.55 | **65.69** ± 4.13 | **70.28** ± 5.02 | 74.33 ± 1.32 | **75.67** ± 0.56 | **84.14** ± 0.45 |

Table 6: Ablation study results. (The number of samples per class is 2)

| Model | Cora | Citeseer | Pubmed | CoauthorCS | Computer | Photo |
|---|---|---|---|---|---|---|
| HDSC-1 | 80.89 ± 2.15 | 71.35 ± 1.56 | 72.12 ± 3.20 | 84.56 ± 0.49 | 81.77 ± 0.93 | 88.21 ± 0.93 |
| HDSC-2 | 81.86 ± 1.77 | 71.89 ± 0.78 | 73.68 ± 2.15 | 85.08 ± 0.76 | 82.66 ± 0.47 | 89.30 ± 0.45 |
| HDSC | **82.50** ± 0.85 | **72.31** ± 1.16 | **75.80** ± 3.82 | **85.35** ± 1.13 | **82.78** ± 0.55 | **89.57** ± 0.77 |

Table 7: Ablation study results. (The number of samples per class is 5)

| Model | Cora | Citeseer | Pubmed | CoauthorCS | Computer | Photo |
|---|---|---|---|---|---|---|
| HDSC-1 | 81.02 ± 2.10 | 71.98 ± 1.57 | 76.55 ± 3.19 | 84.55 ± 0.57 | 82.78 ± 0.61 | 90.23 ± 1.28 |
| HDSC-2 | 82.48 ± 1.57 | 72.54 ± 2.03 | 78.66 ± 2.41 | **85.97** ± 0.83 | 83.08 ± 0.45 | 91.04 ± 0.58 |
| HDSC | **84.34** ± 0.77 | **73.81** ± 0.98 | **82.49** ± 1.04 | 85.85 ± 0.67 | 83.75 ± 0.43 | **91.12** ±0.35 |

Table 8: Ablation study results. (The number of samples per class is 10)

| Model | Cora | Citeseer | Pubmed | CoauthorCS | Computer | Photo |
|---|---|---|---|---|---|---|
| HDSC-1 | 82.64 ± 1.25 | 72.75 ± 1.15 | 78.84 ± 1.20 | 89.32 ± 0.95 | 84.67 ± 0.85 | 92.40 ± 0.58 |
| HDSC-2 | 84.68 ± 1.40 | 73.46 ± 1.05 | 82.89 ± 1.30 | 90.57 ± 0.88 | 85.82 ± 0.72 | 93.30 ± 0.46 |
| HDSC | **85.75** ± 0.91 | **74.63** ± 1.39 | **84.11** ± 0.92 | **91.33** ± 0.45 | **86.48** ± 0.47 | **94.54** ±0.31 |

Table 9: Ablation study results. (The number of samples per class is 20)

**Analysis of $\zeta_{node}$, $\zeta_{edge}$, and $\zeta_{triangle}$.** Figure 3 presents the parameter space reflecting the impact of time memory at different levels $\zeta_{node}$, $\zeta_{edge}$, and $\zeta_{triangle}$ on the model's diffusion performance. As $\zeta_{node}$ increases from 0.7 to 1.0, we observe a significant improvement in model accuracy, indicating that the local information of nodes and their long-term memory play the most crucial role in global predictions. When fixing the node topological memory $\zeta_{node} = 1.0$, as shown in Figures 4 and 5, $\zeta_{edge}$ exhibits diminishing marginal returns, suggesting that excessive reliance on local edge connections weakens the transmission of global information. Regarding $\zeta_{triangle}$, the results show that the role of higher-order structures lies more in complementing local complexities. Therefore, under the hierarchical topological memory mechanism, the structure where nodes dominate, edges play a secondary role, and triangles provide auxiliary support can effectively enhance the diffusion model's adaptability to complex graph structures, thereby improving overall performance.

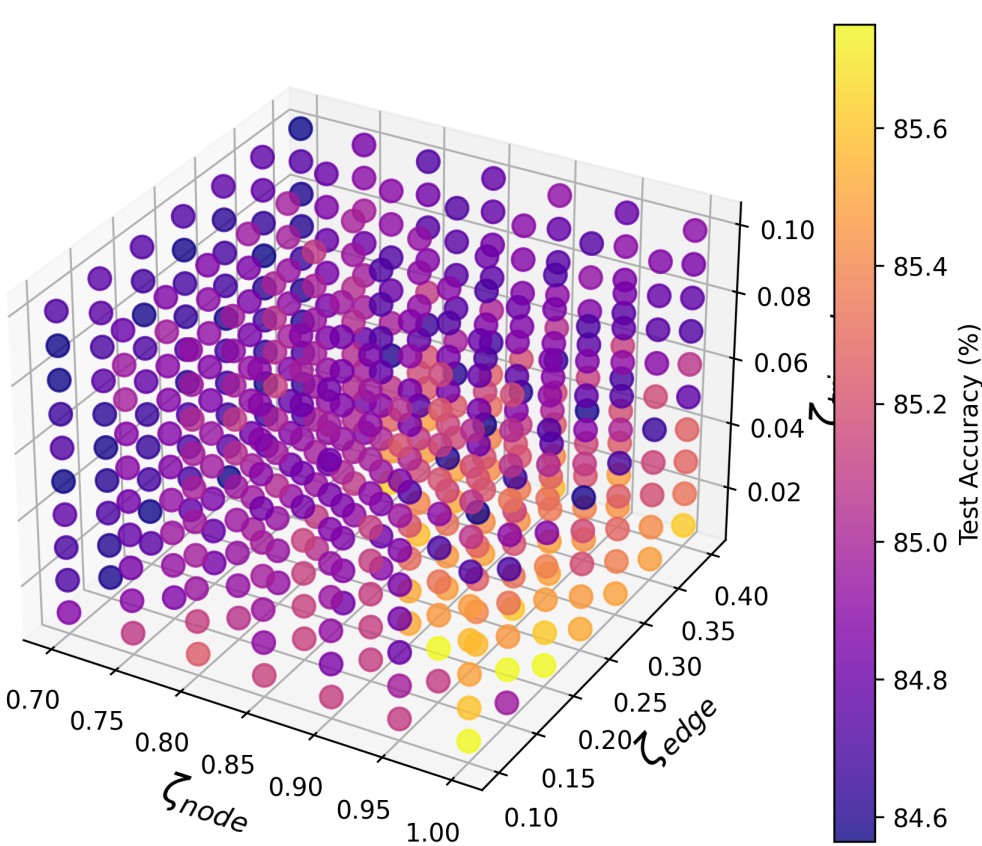

Figure 3: The parameter space of $\zeta_{node}$, $\zeta_{edge}$ and $\zeta_{triangle}$

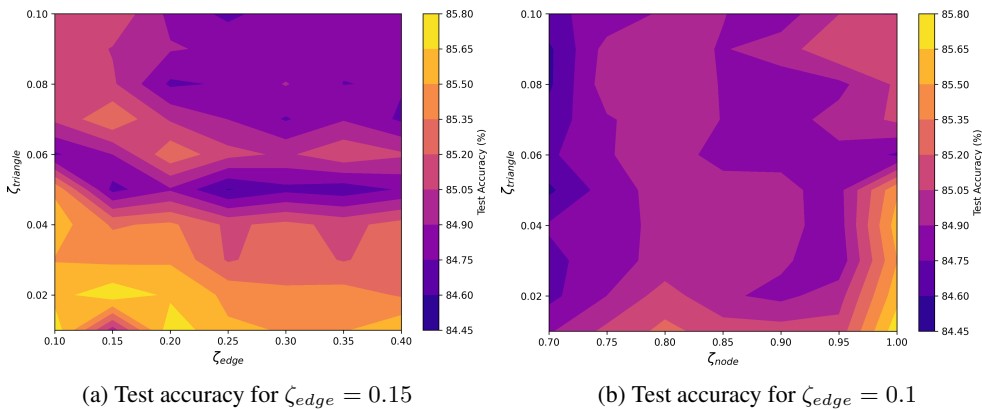

(a) Test accuracy for $\zeta_{edge} = 0.15$

(b) Test accuracy for $\zeta_{edge} = 0.1$

Figure 4: The impact of the relative changes of $\zeta_{node}$ and $\zeta_{triangle}$ when $\zeta_{edge}$ is fixed.

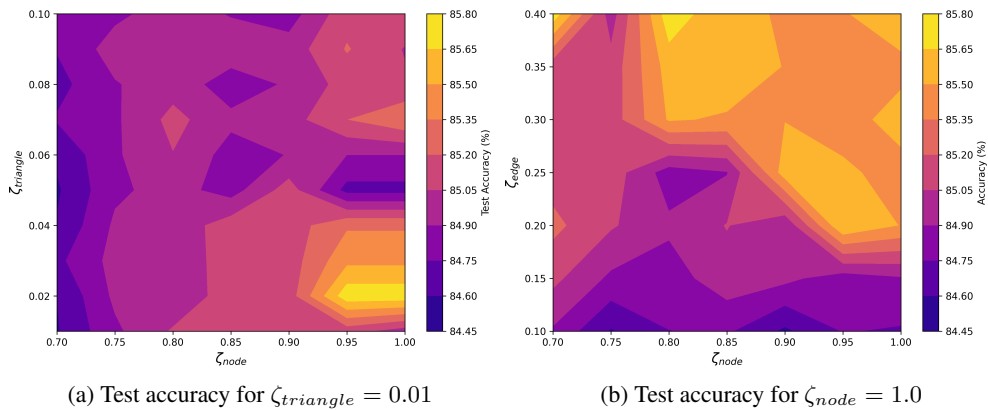

(a) Test accuracy for $\zeta_{triangle} = 0.01$           (b) Test accuracy for $\zeta_{node} = 1.0$

Figure 5: The impact of changes in $\zeta_{edge}$ on the results when $\zeta_{node}$ and $\zeta_{triangle}$ are fixed, respectively.

