# OpenReview forum: "A Unified Framework for Hierarchical Diffusion via Simplicial Complexes"
_ICLR.cc/2025/Conference — Submitted to ICLR 2025_

### Official Review · Reviewer_DGUR · 2024-11-02

**Soundness:** 1
**Presentation:** 2
**Contribution:** 2
**Rating:** 3
**Confidence:** 5

**Summary:**

The paper proposes a hierarchical diffusion framework (HDSC) using simplicial complexes to enable adaptive diffusion across different structural levels in graphs (nodes, edges, triangles). It introduces a time-dependent topological memory mechanism and demonstrates performance improvements on standard graph benchmarks.

**Strengths:**

Manuscript presents an elegant formulation of hierarchical diffusion using simplicial complexes. I liked the attempt in integration of memory mechanisms with topological structures. The proposed approach can offer a systematic approach to multi-level information propagation. Using the boundary operators for connecting different structural levels can be interesting, and the experimental validation on multiple datasets seems promising.

**Weaknesses:**

The literature review can be improved, currently
- The progression from GNNs to diffusion-based approaches is incoherrent
- Unclear positioning relative to existing higher-order methods
- Poor articulation of connections between cited works

The simplicial complex features are unclear, for instance triangle features appear to be mere statistical aggregations of lower-order features as there is nothing noted in the manuscript about how the authors did that and this is based on my best guess.
The performance improvements might reflect better feature aggregation rather than true higher-order structural learning as the benchmark datasets don't demonstrate genuine higher-order interactions. Given that, it is unclear that how the computational complexity justifies marginal improvements. Even with that, the scalability is limited beyond triangles in Ω^(t).

Memory mechanism appears arbitrarily borrowed from LSTM/GRU without justification
Complex hyperparameter landscape without clear tuning strategy

In the results section, I wished for a comparison with simpler feature aggregation methods done on datasets with genuine higher-order features. Again, analysis of computational overhead is necessary.

I found the introduction of topological constraint (Equation 4: B_k ∘ B_k−1 = 0) interesting, but my enthusiast damped down significanlt when I did not find its use in the subsequent development. How does this important topological consistency property is maintained during learning?

**Questions:**

How do authors justify that the method captures genuine higher-order information rather than just better feature aggregation?
Why were standard graph benchmarks chosen instead of domains with inherent higher-order features (e.g., protein complexes, neural synchronization)?
What motivated the specific choice of memory mechanism?
How does the framework extend beyond triangles?
How practical is the hyperparameter tuning in real applications, given the multiplication of them in (18) and (19).
What is the computational overhead compared to simpler methods?
How does the method scale with increasing order of simplicial complexes?
Can similar performance be achieved with simpler feature aggregation approaches? Focusing on short cycles only?
How does this important topological consistency property in (4) is maintained during learning?

---

### Official Review · Reviewer_aRQL · 2024-11-03

**Soundness:** 2
**Presentation:** 1
**Contribution:** 1
**Rating:** 3
**Confidence:** 5

**Summary:**

The paper proposes a label propagation model on graphs that combines a form of LSTM-inspired memory mechanism and takes into account higher-order cliques (triangles, etc.). Experiments are done on popular graph datasets and against state-of-the-art graph neural network models.

**Strengths:**

The idea of using an LSTM-like mechanism to mitigate overshooting in graph neural networks is interesting. Moreover, incorporating higher-order structures directly within the label diffusion process is a sensible approach, albeit not entirely novel.

**Weaknesses:**

- The paper is challenging to read due to inconsistent notation and terminology throughout (see questions for some examples).
- Although the discussion centers on simplicial complexes, the approach appears limited to graphs where triangles are treated as simplices. Notably, the experiments are conducted solely on graph data.
- The rationale for restricting to triangles (as in Equation 19) is unclear, with no justification provided for omitting higher-order structures such as 4-cliques and beyond.
- The experimental baselines rely on graph structures alone, while the proposed model explicitly leverages triangle information. This comparison lacks fairness; baselines should also be adapted to consider triangles, such as through clique-expanded graphs and the corresponding adjacency. Additionally, comparisons with available models that directly handle hypergraphs or higher-order structures should be implemented.

**Questions:**

- why did you choose to cite  (Qureshi et al., 2023; Giraldo et al., 2023; Chen et al., 2023) as a reference to the over smoothing problem in guns?
- the term "diffusion models" is not conventional in this context, as it usually refers to generative denoising processes
- L_k is used in (3), (6), etc but not defined
- why do you use a $\circ$ in equation (4) and not elsewhere?
- in line 191 you mention looking at equation (3) under the limit $\Delta t\to 0$. Isn't that simply eq (1)?
- L_1 defined in eq (8) and L_2 defined in eq (10) are quite unusual, and in particular, they are not the kth Hodge Laplacians. This choice should be justified better.
- I am not sure why you need both eq (12) and (13) as one is the Euler discretization of the other, and I am not sure why your motivation is Simplicial Complexes and Topology, but then you don't use Hodge Laplacians.
- In Algorithm 1: you start with a graph while the story until now assumes you have a simplicial complex; the terms $\Phi_nmu$ etc on line 6 do not appear in equation (17); your statement "use the ADAM optimizer to update the loss" is not clear as you are now using the entire data matrix $X$ and higher-order graph interaction, thus it is not obvious how the batching process can be done efficiently
- what is the rationale for restricting to triangles?
- The experiments are not fair, as discussed above

---

### Official Review · Reviewer_MZY1 · 2024-11-03

**Soundness:** 2
**Presentation:** 1
**Contribution:** 3
**Rating:** 1
**Confidence:** 4

**Summary:**

The work studies a method for classification of nodes in a graph, that will rely on diffusion of  features not only between nodes, but on the higher-order structures known as simplicial complexes. Specifically, the work considers nodes, edges and triangles and exhibit the diffusion processes at these different levels (which act as different levels of description of the graphs). The proposition is to use this multi-level diffusion to propagate information in the graph. Then, the authors consider a recurrent architecture, with a memory of how these diffusions are done along the time and there is a trade-off at each step between actual diffusion and the memory of previous diffusions. This provides an algorithm to train the model of diffusion with recurrent memory, so that it can solve a task of supervised classification of nodes, using a classical cross-entropy as loss function. Some experiments are conducted on classical datasets, with expected benchmarks. The impact of the trade-off mentioned above is also studied.

**Strengths:**

The interesting point of the article is the idea to consider diffusions at various levels of the simplicial complexes in the graphs and then combine them with a memory mechanism to provide a new architecture for supervised node classification.

**Weaknesses:**

I group the main weaknesses of these articles under two sets:

1) There are wrong statements (are statements I don't understand given the writing and the mathematics in the artcile) in the presentations, and there miss some elements, particularly in Section 3:

- I question the notations used in 3.1 and the soundness of the derivations from eq. (6) to eq. (11). Note that I guess that the authors used eq. (13) which is correct. But not for the reasons stated previously.
- One takes usually the boundary operators B_k in the transpose meaning, e.g. for B_1, or size N per |E|. See for instance Schaub et al., "Signal processing on higher-order networks: Livin’ on the edge... and beyond" 2021, for notations that are customary
- One should not mix up continuous time and discrete time notations. This is confusion and unnecessary as, in the end, the authors use only eq. (18) in discrete time.
- Eq (9) is wrong: N_2 should be a set of edges (not of triangles) and B_1^T can only be applied to features on edges, not on triangles
- The authors should explain where are the features ? In the graphs of the numerical examples, there are initially only features on nodes; how are the features on other simplicial complexes obtained ? Are they gradients (on edges), curls (in triangles),... ? Do they have some initial value ?
- Why suddenly go to normalized boundaries operators in (12) ?

Globally, all section 3.1 has to be reconsidered in a rigorous way.

- For 3.2, the proposition appears interesting but, as I reader, I have no insight about why this choice of architecture, nor about the trade-off between diffusion and memory, about the relative weights of the nodes / edges / triangles terms.

Here again, 3.2 should be written in a more comprehensive an thoughtful way.


2) Despite some relevant ideas, the method is not really impressive and is currently limited to a task (node classification, tested on graphs of small sizes). There are also questionable choices for presentation:

- The task studied is only specified on line 376. Node classification being also an instance of semi-supervised learning, comparisons to SSL methods can be expected.
Also, on this task and for the methods considered here, one sees saturation of performance on the datasets used. Why these choices of datasets and of task ?

- The datasets used are small scale, have been used since many years (the most recent is from 2018) and given the considerable number of articles and larger scale benchmarks existing now, one questions the choice of limiting the study to small graphs.

- The use of a  train / test / validate procedure is not clear. It's not clear whether some nodes were reserved for them never to be seen during training nor testing.

- The study about the impact of the choice of $K$ is not enough, only in Appendix B and with no insight nor theoretical element.

- What is (are) the method(s) for Visualization in 4 ?

- I would not say that "for most cases, HDSC significantly outperforms other baseline models across six datasets" on line 377.
Performance are only slightly above... And I have doubts about the validation given that the choices of the hyperparameters  (in Table 4) change from one graph to another.

**Questions:**

* Many questions or suggestions or re-writing are already above in the part about "Weaknesses".


* The architecture for "Topological Memory" of section 3.2 is very reminiscent of the RNN methods, especially the GRU architecture. The authors should compare their proposition to existing RNN methods so as to state and reference correctly previous works.

* There study about the impact of the choice of $K$ is not enough. I think I understood that the authors take $K=2$ to consider nodes, edges and triangles but the actually important question is to understand what orders are useful or necessary ? And to find a way to balance them. Some elements are reported in Appendix B but they are only numerical and not discussed in the main text, while this question is one of the most relevant and important about the present method.

---

### Official Review · Reviewer_VR72 · 2024-11-09

**Soundness:** 3
**Presentation:** 3
**Contribution:** 4
**Rating:** 8
**Confidence:** 4

**Summary:**

The paper proposes a new framework to pass around the feature representation of the nodes in a graph. Existing work introduces a diffusion process to facilitate the exchange of information. However, it introduces new problems such as node feature homogenization. The new technique involves identifying the higher-order structures, i.e., simplicial complexes, on the graph. And then, the authors introduce (1) a new diffusion mechanism among the simplices, and (2) a new memory mechanism to allow richer information retention. Eventually, they successfully extended the conventional framework and provided persuasive experimental results.

**Strengths:**

This is a great paper that moves the field forward by (1) extending existing approaches in diffusion process modeling in GNNs to higher-order structures, and by (2) incorporating a novel topological memory mechanism, they enrich the framework with new model structures, and finally (3) achieving better performance over traditional methods. I think the paper is well-written. The authors may consider to address some of the notational questions that I will raise later.

**Weaknesses:**

The many baselines confuse me. I do not understand the differences among the six other baselines, except for that they are of either plain GNN models, or graph diffusion-based models. The authors might consider highlight their differences in Section 1, for example. in Paragraphs 1 and 2. And then provide a hint to the reader that you will compare these certain methods in Section 4.

**Questions:**

Line 157: The definition about the k-th-level Laplacian is missing. (In math, they are called "Hodge Laplacians." See https://www.stat.uchicago.edu/~lekheng/work/psapm.pdf.) It might be helpful if they can be defined in terms of the boundary operators. My comment: The Hodge decomposition (see https://epubs.siam.org/doi/10.1137/18M1201019) may allow future analysis and extension of the proposed framework.

Lines 162–176, "Topological consistency":
[1] My understanding for the boundary operator is to connect the entities between adjacent k-simplicial spaces. Therefore, B_k should belong to R^{n, m} where m is the number of k-simplices, and n is the number of (k-1)-simplices. Can the authors clarify this point?
[2] Separately, Eq. (4) should also be B_(k-1) \circ B_k = 0, because we apply the operator from left to right. The definition is standard in algebraic topology, in that we allow some information in a higher-dim to disappear when we transfer to lower ones.
[3] A consequence of this definition is that the kernel of B_(k-1) is same as the image of B_k. It would be nice if the authors can explain why the property is desirable.
[4] Finally, Eq. (5) needs some elaboration about what the authors mean by "consistency."

Comment: You mentioned "efficient coupling between different structural levels" several times. But I think only "adjacent levels" are coupled. Could you clarify? If it's indeed the case, I suggest some Find+ReplaceAll for clarity.

Misc question 1: Is there a reason why you only include Chamberlain et al., 2021 and Thorpe et al., 2022 in the experiments? But not Liu et al., 2024 and others, which are mentioned in Section 1.

Misc question 2: How do you define the higher-order simplices? Do you simply identify a k-connected complete subgraph as a (k-1)-simplex, in a flag or clique complex fashion? If so, is it a good idea to mention it in the paper?

---

### Meta-Review · Area_Chair_qLWe · 2024-12-19

**Metareview:**

The authors propose a node classification method that diffuses node features across various higher-order structures, specifically nodes, edges, and triangles. The approach incorporates a memory module to retain and update diffused features over time. Experiments on six real-world datasets demonstrate the competitive performance of the proposed method.

All reviewers found the ideas of multi-level diffusion and the memory module interesting and promising, and they acknowledged the empirical effectiveness of the proposed method.

However, most reviewers raised the following common concerns:
- The experiments are limited to datasets lacking genuine higher-order structures, and the baselines do not account for such structures, despite the availability of datasets (e.g., hypergraphs) and baselines that do.
- The exploration of even higher-order structures, such as 4-cliques, has been overlooked.
- Specific design choices, particularly the introduction and structure of the memory module, require stronger motivation and justification.
- The presentation and methodological rigor leave significant room for improvement.

The paper presents interesting and promising ideas, but the current version is not yet ready for publication; therefore, the meta-reviewer recommends rejection.

**Additional Comments On Reviewer Discussion:**

The authors did not provide rebuttals.

---

### Decision · Program_Chairs · 2025-01-22

Reject